# Diurnal Changes in Water Soluble Carbohydrate Components in Leaves and Sucrose Associated *TaSUT1* Gene Expression during Grain Development in Wheat

**DOI:** 10.3390/ijms21218276

**Published:** 2020-11-05

**Authors:** Sarah Al-Sheikh Ahmed, Jingjuan Zhang, Hussein Farhan, Yingquan Zhang, Zitong Yu, Shahidul Islam, Jiansheng Chen, Sanda Cricelli, Andrew Foreman, Wim Van den Ende, Wujun Ma, Bernard Dell

**Affiliations:** 1Agricultural Sciences, College of Science, Health, Engineering and Education, Murdoch University, 90 South Street, Murdoch, WA 6150, Australia; S.Al-SheikhAhmed@murdoch.edu.au (S.A.-S.A.); husseinfarhan92@yahoo.com.au (H.F.); zhangyingquan821@126.com (Y.Z.); zitongyu@outlook.com (Z.Y.); S.Islam@murdoch.edu.au (S.I.); jshch@sdau.edu.cn (J.C.); W.Ma@murdoch.edu.au (W.M.); 2College of Science, Health, Engineering and Education, Murdoch University, 90 South Street, Murdoch 6150, WA, Australia; S.Cricelli@murdoch.edu.au (S.C.); A.Foreman@murdoch.edu.au (A.F.); 3Laboratory of Molecular Plant Biology, 3001 Leuven, Belgium; wim.vandenende@kuleuven.be

**Keywords:** diurnal pattern, gene expression, grain filling, sucrose transporter (SUT), water soluble carbohydrate (WSC) remobilization, wheat

## Abstract

In plant tissues, sugar levels are determined by the balance between sugar import, export, and sugar synthesis. So far, water soluble carbohydrate (WSC) dynamics have not been investigated in a diurnal context in wheat stems as compared to the dynamics in flag leaves during the terminal phases of grain filling. Here, we filled this research gap and tested the hypothesis that WSC dynamics interlink with gene expression of *TaSUT1.* The main stems and flag leaves of two genotypes, Westonia and Kauz, were sampled at four hourly intervals over a 24 h period at six developmental stages from heading to 28 DAA (days after anthesis). The total levels of WSC and WSC components were measured, and *TaSUT1* gene expression was quantified at 21 DAA. On average, the total WSC and fructan levels in the stems were double those in the flag leaves. In both cultivars, diurnal patterns in the total WSC and sucrose were detected in leaves across all developmental stages, but not for the fructans 6-kestose and bifurcose. However, in stems, diurnal patterns of the total WSC and fructan were only found at anthesis in Kauz. The different levels of WSC and WSC components between Westonia and Kauz are likely associated with leaf chlorophyll levels and fructan degradation, especially 6-kestose degradation. High correlation between levels of *TaSUT1* expression and sucrose in leaves indicated that *TaSUT1* expression is likely to be influenced by the level of sucrose in leaves, and the combination of high levels of *TaSUT1* expression and sucrose in Kauz may contribute to its high grain yield under well-watered conditions.

## 1. Introduction

Improving and sustaining grain yield remains a priority for wheat production as the world population increases [1]. The major phenotypic contributors to wheat grain yield are thousand grain weight (TGW) and kernel number per spike (KN) [2,3,4]. The sum of glucose, fructose, sucrose and fructans, hereafter termed “water soluble carbohydrate” (WSC) in wheat stems and leaf sheaths is one of the crucial parameters contributing to grain yield especially under terminal drought conditions [5,6,7,8,9]. Interestingly, the remobilization of stem WSC is subject of considerable genetic variation, and molecular marker screening has been useful in identifying suitable lines for pre-breeding [7].

During photosynthesis in plants, carbon dioxide is assimilated into triose phosphates in the Calvin cycle in chloroplasts, from which sucrose is formed and exported to an array of sinks [10]. Photosynthesis from leaves, sheath, culm, and ears contributes to the total photosynthetic capacity in wheat [11]. Among those, flag leaf area photosynthesis correlates well with the grain yield in wheat [12]. Chlorophyll is an important photosynthetic pigment largely contributing photosynthetic capacity [13]. Chlorophyll fluorescence provides a signature of photosynthesis and can easily be measured with portable chlorophyll fluorometers [14].

Sucrose is the major transportable sugar in plants [15]. The main enzymes involved in leaf sucrose synthesis include cytosolic fructose 1,6-bisphosphatase, glucose 6-phosphate isomerase (phosphoglucose isomerase), phosphoglucomutase, uridine diphosphate-glucose pyrophosphorylase, sucrose phosphate synthase (SPS) and sucrose phosphate phosphatase [15]. The activity of SPS correlates well with photosynthetic rates. However, when exogenous sucrose is supplied, SPS activities decrease and photosynthesis is affected [15,16,17]. Excess sucrose can lead to vacuolar fructan accumulation in fructan plants such as wheat [8]. Sucrose is acting as a signaling molecule in mediating source-sink relationship in plants [18]. For instance, exogenous sucrose reduced the impact of heat-stress on spikelet fertility, kernel weight, and dry weight at the flowering stage in rice [19]. The role of sucrose as a signaling molecule in plants may involve interactions with endogenous plant hormones, such as abscisic acid (ABA) as it was reported that ABA enhances sugar metabolism and transport into spikelets of rice, preventing pollen abortion by increasing the expression levels of *SUT1*, invertase (*INV*), and sucrose synthase (*SUS*) genes at the meiosis stage in pollen mother cells [20]. Both exogenous sucrose and ABA enhanced the starch contents and the enzyme activities involved in starch synthesis in rice grains [21].

Fructan is the main component of wheat stem WSC, comprising 80% of the WSC in internodes [5,8,22,23,24]. Fructans are polysaccharides consisting of fructosyl residues and a terminal glucose residue. In wheat, branched graminan- and linear levan-type fructans (a majority of *β*-(2-6) linkages) predominate. They are synthesized by the action of enzymes sucrose:sucrose 1-fructosyltransferase (1-SST), fructan 6-fructosyltransferase (6-SFT), and fructan 1-fructosyltransferase (1-FFT) [25,26]. Fructans are degraded into sucrose and fructose by specific fructan exohydrolases (FEH) during WSC remobilization while sucrose is hydrolyzed to hexoses by different types of invertases (INVs) [27,28,29].

Plant carbon resource allocation changes considerably throughout developmental stages and during stress responses, but also requires a very flexible fine tuning throughout the diurnal cycle, orchestrating leaf starch dynamics during the light and dark periods to the carbon demands of sink organs [28,29,30,31]. The metabolite diurnal changes in leaves affect both total carbon gain and the growth of leaf and root [30,31,32]. The biological clock and sugar signaling fulfil critical roles in such processes [33] with trehalose 6-phosphate signaling emerging as an important regulator [34].

In source leaves of maize and citrus plants, the levels of glucose, sucrose, and starch displayed diurnal patterns, with maximum accumulation through the day and minimum at night [35,36,37]. Diurnal patterns of sugar transport in plants are important as (1) sugar turnover in leaves prevents the potential suppression of photosynthesis, enhancing total carbon gain, (2) carbon needs to be supplied to sink tissues stimulating their growth, and (3) stored carbon should be mobilized to sustain growth and maintenance at night. So far, leaves have been the main focus to study carbohydrate dynamics in a diurnal context. Although dynamic fluxes of wheat stem and root WSC components have been investigated throughout the period of wheat grain filling [8,9], to the best of our knowledge no studies have been performed yet to study the carbohydrate dynamics in these organs in a diurnal context. Here, we aim to fill this knowledge gap in the context of wheat grain filling. However, it is clear that sugar levels may greatly depend on the sugar transport activities [15,38] presenting in these tissues, and how this may vary in a diurnal context.

Sucrose levels in maize leaves reflected diurnal patterns of *ZmSUT1* gene transcript levels [36]. Furthermore, *ZmSUT2* also showed diurnal patterns of gene expression in maize [38]. The diurnal patterns of *ZmSut2* gene expression in maize source and sink leaves inspired us to investigate whether diurnal changes also occur in wheat stem and leaf WSC, and whether sucrose levels could correlate with *TaSUT1* gene expression. Therefore, the objective of this study is to identify whether the differences in diurnal changes of WSC components can be explained by different *TaSUT1* expression levels occurring in different genotypic backgrounds. The overall hypothesis is that sucrose transporter expression and tissue concentrations of all WSC components may be affected by the biological clock, but in a variable way depending on developmental stages and plant genotypes.

## 2. Results

### 2.1. Grain Weight

Across the developmental stages, grain weight was recorded for the stems used for sugar measurements. The grain weight in Kauz accumulated faster than in Westonia between 21 and 28 days after anthesis (DAA), resulting in significantly higher grain weight per spike in Kauz at 35 DAA (Figure 1a).

At the final harvest, the kernel number per spike (KN) of Kauz (57) was significantly higher than Westonia (44) (*p* < 0.01) (Figure 1b) while the TGW were similar. Also, the grain weight per spike of Kauz (2.6 g) was significantly higher than that of Westonia (2.0 g). The differences in biomass, tiller number, and plant height were not significant. The higher grain weight per spike in Kauz was mainly due to the higher KN.

### 2.2. Leaf Area and Leaf Senescence

The average leaf area of the flag leaves (sampled on a daily basis) were used for presenting the flag leaf area patterns across developmental stages (Appendix A). The flag leaf area grandually expanded from heading until 7 and 14 DAA in Kauz and Westonia, respectively. After a drop at 21 DAA, the values of the flag leaf area returned to previous levels at 28 DAA in both varieties. Overall, the flag leaf area of Westonia was significantly higher than in Kauz (Appendix A). No diurnal changes were identified in the flag leaf area of either variety (Appendix A).

The chlorophyll index showed that senescence in Kauz commenced earlier than in Westonia. In both the flag and the second leaves, the chlorophyll content of Kauz was higher before 14 DAA and then declined (Figure 2). By contrast, Westonia stayed green until the late stages of grain filling (28 and 35 DAA).

### 2.3. Concentrations and Diurnal Patterns of WSC and Fructan

#### 2.3.1. The Total WSC

The daily average of the total WSC levels of the flag leaves and stems were used for presenting the WSC patterns during developmental stages. Trends in the levels of the total WSC in the flag leaves and stems are shown in Figure 3a. On average, the WSC levels in the flag leaves were about half of that in the stems. The WSC levels in the flag leaves of both varieties increased gradually from 5.7% at anthesis to 11.2% at 14 DAA and then plateaud, while in the stems, it increased steeply from 13 to 30%, then declined from 21 DAA. Compared with Westonia, the WSC levels in Kauz increased faster after anthesis in both leaves and stems while the increase in Westonia was delayed until 14 DAA. Compared with the peak levels of the flag leaves (11.2% WSC), the significantly higher levels of stem WSC (30%) demonstrated stem storage function.

The diurnal WSC in flag leaves and stems were analysed at four hourly intervals for six weeks during the stages of grain development (Figure 3b). There were more obvious diurnal patterns of WSC concentrations in leaves than in stems. High levels of leaf WSC appeared in the day (08:30 to 20:30) at all stages of grain development, while in the stems, the diurnal pattern was only clear in Kauz at anthesis (0 DAA). In general, stem WSC levels remained stable over the 24 h period. The diurnal pattern of stem WSC in Kauz at anthesis suggests high energy requirement during the night at this stage of development in Kauz.

The results showed the leaf WSC levels accumulated during photosynthesis through the day and were remobilized under darkness. Compared with Westonia, the increment in WSC in the flag leaves of Kauz rose earlier during the day at earlier stages (up to 14 DAA) whereas the reverse occurred at 28 DAA. The high levels of the leaf WSC in Kauz reflect high levels of the stem WSC before 14 DAA. Notably, the stem WSC concentrations in Westonia were significantly higher than that in Kauz at 21 and 28 DAA.

#### 2.3.2. Fructan

The daily average fructan levels of the flag leaves and stems were used for presenting the fructan patterns during developmental stages. Appendix A shows that the fructan levels in both the flag leaves and stems are similar to the total WSC throughout all developmental stages. Comparing to WSC, at the maximum, the fructan level in the flag leaves counted 59% of the WSC and it was 83% of the WSC in the stems. On average, the stem fructan concentration was 2.4 times higher than that in the flag leaves.

Unlike leaf WSC, diurnal patterns of leaf fructan were only present at heading, and 14 and 21 DAA in both varieties. Like the stem WSC, there was a diurnal pattern in stem fructan only in Kauz at anthesis. The remobilization of fructan at night in Kauz indicates the energy requirement at anthesis (Appendix A).

### 2.4. Concentrations and Diurnal Patterns of Glucose and Fructose

#### 2.4.1. Glucose

The daily average glucose levels of the flag leaves and stems were used to illustrate the glucose patterns over the developmental stages. The leaf glucose levels (1–l2.2%) were slightly lower than those in the stems (1.3–2.7%). Leaf glucose increased steadily in Westonia across the developmental stages but there were no significant differences in Kauz. The stem glucose levels in Kauz were markedly higher at heading and anthesis and gradually decreased to a steady low-level during grain filling. The stem glucose levels in Westonia slowly increased until 7 DAA and then progressively decreased to a similar low-level as in Kauz (Figure 4a).

Significant diurnal patterns of leaf glucose levels were present at heading, anthesis, and 7 and 21 DAA in both varieties. There were no leaf diurnal patterns at 28 DAA for glucose in either variety. In the stems, the glucose concentrations in Kauz were consistently higher at heading during the day and lower at 14 DAA (Figure 4b). No diurnal patterns were present in stem glucose levels for both varieties (Appendix A).

#### 2.4.2. Fructose

In general, the daily average fructose concentrations of the flag leaves and stems were similar, and the levels were within the range of 1.4 to 3.5% across developmental stages. In the flag leaves, and similar to glucose patterns, the fructose concentrations of Westonia gradually increased from 1.4% to a significant level of 3% while it was relatively stable in Kauz. In the stems, the fructose concentrations of Kauz remained at a similar level (1.9–2.7%) until 14 DAA and then increased significantly to 3.5%, while the fructose concentrations in Westonia were lower at 14 and 21 DAA compared with other developmental stages (Figure 5a).

Like glucose, leaf fructose diurnal patterns were clearly present at heading, anthesis, and 7 and 21 DAA in both varieties. Significantly high fructose concentrations occurred at 12:30 and 16:30 during the day before 14 DAA. There was another peak at 00:30 for both varieties at 14 DAA and for Westonia at 28 DAA (Figure 5b). There were no diurnal fructose patterns in the stems across all developmental stages (Appendix A). Stem fructose levels in Westonia were significantly higher than that in Kauz at 7 DAA and the opposite occurred at 21 and 28 DAA.

### 2.5. Concentrations and Diurnal Patterns of Sucrose

The sucrose patterns during developmental stages showed that the leaf sucrose level was slightly lower than in the stems. Leaf sucrose concentrations in Kauz increased steeply at 14 DAA and then plateaued (2.4%) whereas, in Westonia, concentrations increased after 21 DAA to reach 2.4%. In the stems of both varieties, sucrose levels increased gradually until 21 (3.5%) and 28 DAA (2.9%) for Kauz and Westonia, respectively. The stem sucrose concentrations in Kauz were significantly higher than those in Westonia before 28 DAA while the sucrose concentrations were only high at 21 DAA in the flag leaves of Kauz (Figure 6a).

The diurnal patterns of sucrose concentrations in the flag leaves were present at all developmental stages for Kauz, and at 14, 21, and 28 DAA in Westonia. The diurnal patterns illustrated the accumulation of sucrose during the day between 08:30 to 16:30 and decrease at night in the flag leaves of both varieties (Figure 6b). There was no diurnal sucrose pattern in the stems of either variety (Appendix A).

### 2.6. Concentrations and Diurnal Patterns of 1-Kestose, 6-Kestose and Bifurcose

#### 2.6.1. 1-Kestose

The daily average 1-kestose levels of the flag leaves and stems were used for presenting the 1-kestose patterns during developmental stages. On average, the 1-kestose levels in the leaves were one-third of that in the stems. The levels of 1-kestose in the flag leaves were similar between Westonia and Kauz except for the final point, when it was significantly higher in Kauz (0.16%) than Westonia (0.10%). In the stems, 1-kestose levels in Kauz were much higher than those in Westonia except at the stages of 14 and 28 DAA (Appendix A).

Through diurnal recording, the 1-kestose levels in the flag leaves were close to zero at heading, anthesis, and 7 DAA. The leaf diurnal patterns of 1-kestose were present at 14 DAA for both varieties. No significant diurnal patterns were observed at 21 and 28 DAA in flag leaves and no diurnal patterns were detected at all stages in stems (Appendix A).

#### 2.6.2. 6-Kestose

The concentrations of 6-kestose in the stems (5.1%) were 10 times higher than those in the flag leaves (0.5%). In leaves, the 6-kestose concentration in Westonia was significantly higher than in Kauz at 28 DAA. In the stems of Kauz, 6-kestose concentrations were strongly enhanced after heading until 14 DAA, and then decreased sharply to 0.06%; while in Westonia, the 6-kestose levels increased after anthesis until 14 DAA and then remained constant. Compared to Westonia, the 6-kestose concentrations in Kauz were markedly higher before 14 DAA while it was significantly lower after that stage (Figure 7a).

There was no significant diurnal pattern of 6-kestose in leaf and stem tissues for either variety (Appendix A), except for a slight increase at 28 DAA in stems at 20:30 in Westonia.

#### 2.6.3. Bifurcose

The higher stem bifurcose concentrations were about six times those of the flag leaves. Similar to 6-kestose in the flag leaves, the bifurcose concentration of Westonia (0.13%) in flag leaves was significantly higher than that of Kauz (0.07%) at 28 DAA. In stems, the bifurcose concentrations in Kauz increased earlier until 14 DAA and then dropped while in Westonia, they increased more slowly until peaking at 14 DAA, then decreased sharply until 21 DAA and remained at constant low concentration to 28 DAA. Bifurcose concentrations in Kauz stem were significantly higher than in Westonia before 14 DAA but the reverse occurred at 28 DAA (Figure 7b).

Like 6-kestose, there were no significant diurnal patterns of bifurcose levels in the flag leaves and stems for both varieties (Appendix A).

### 2.7. TaSUT1 Gene Expression in the Flag Leaves and Stems at 21 DAA

Sucrose transporters (SUT) play a role in the active transport of sucrose across the membrane combined with proton transport [39]. In comparison to *TaSUT2, TaSUT3, TaSUT4,* and *TaSUT5* during grain filling, *TaSUT1* was identified as the major gene group with the highest expression level in the sucrose transporter family in wheat [40]. Since noticeable differences in sucrose levels between varieties were present at 21 DAA in both the flag leaves and stems, diurnal *TaSUT1* gene expression was examined at this time of grain development.

The diurnal patterns of *TaSUT1* gene expression in stems were more prominent than in the flag leaves. The *TaSUT1* gene was highly expressed from 12:30 to 16:30, while the expression levels were lower at night in both the flag leaves and stems. At 16:30, the peak values were 22.6 and 31.6 units in the flag leaves of Westonia and Kauz, respectively, whereas in stems, they were 38.0 and 49.8 units, respectively. Dawn was at 06:22 and dusk was at 18:05 at 21 DAA (Figure 8a).

The *TaSUT1* gene expression reflected the levels of sucrose in both the flag leaves and stems. The expression was higher in Kauz than Westonia in the flag leaves and stems. On average, the sucrose concentrations in stems were 18% higher than that in the flag leaves, and accordingly, *TaSUT1* expression levels were 40% higher in the stems compared with the flag leaves (Figure 8a). Compared with Westonia, *TaSUT1* in Kauz was expressed 59 and 77% more in the flag leaves and stems, respectively (Figure 8a). The significant positive correlations between levels of sucrose and *TaSUT1* expression were observed in leaves of Westonia (R^2^ = 0.37 **, *p* < 0.01) and Kauz (R^2^ = 0.47 **, *p* < 0.01) demonstrating the strong linkage between leaf sucrose levels on the expression of this gene (Figure 8b). However, no significant correlations were detected between stem sucrose levels and stem *TaSUT1* expression (Figure 8b).

## 3. Discussion

### 3.1. Differences in Diurnal Patterns of WSC and WSC Components in the Flag Leaves and Stems

To better understand the WSC accumulation and remobilization in wheat, the daily time course of total WSC, and the different WSC components in the flag leaves and stems were considered. Diurnal patterns of total WSC and sucrose were detected in the flag leaves across all developmental stages studied except for sucrose in Westonia at heading. Glucose and fructose levels showed diurnal patterns mainly before 7 DAA. Diurnal patterns in total fructan were present at heading and 14 DAA in both varieties, and at 21 DAA in Kauz. No diurnal pattern was noticed for 6-kestose and bifurcose. In the stems, diurnal patterns in WSC and fructan concentrations were present only in Kauz at anthesis. These results further validate different functions in carbon fixation and storage between the leaves and stems. The WSC is manufactured in leaves during the day and remobilized (or partially used for respiration in situ) under darkness. On average, the stem WSC levels were more than twice those in the flag leaves, illustrating the WSC storage function in the stem in wheat. In maize, diurnal patterns of sucrose and starch concentrations were present in leaves at silking and at 21 days after silking [41]. Diurnal patterns of glucose, fructose and sucrose levels were also observed in three-month-old citrus plants grown hydroponically in a complete nutrient solution in a culture chamber [37].

The diurnal patterns of stem WSC and fructan in Kauz at anthesis indicate high energy consumption at anthesis. The energy requirement in Kauz at anthesis may relate to the high floret number per spike and high fertility rate leading to the significantly higher seed number per spike. In agreement with our previous studies, Kauz constantly showed significantly high kernel number per spike compared with Westonia [4,40].

### 3.2. Associations between Leaf Senescence and WSC

Generally, there is a positive correlation between photosynthetic capacity (*A*m) and specific leaf area [42]. Later, researchers identified that the relationship of leaf area and plant biomass was not linear but varied depending on carbon partitioning [43]. In addition, leaf thickness is a factor in carbon partitioning as thicker leaves have greater capacity to fix carbon than thin leaves [44,45]. Although the flag leaf area of Westonia was significantly higher than in Kauz, this may not necessarily indicate a higher photosynthetic capacity in Westonia. Our previous glasshouse study showed that the rate of photosynthesis was slightly lower in Westonia than in Kauz [6]. It is not known yet whether Kauz leaves are thicker than Westonia. In a *Rht*-near-isogenic line study, the dwarf mutant *Rht-B1c* plants had thicker leaves, less leaf area, and greater drought tolerance compared to the wild type allele *Rht-B1a* [46,47]. It has been suggested that the *Rht-B1c*-encoded DELLA proteins provide some protective mechanism in wheat [48]. Whether dwarf allele *Rht-B1b* would also lead thicker leaves in Kauz [4] is an interesting subject for further investigation. Moreover, the second leaf is important for carbon export, but we did not measure the area of these leaves. The current results showed that the higher flag leaf area in Westonia did not lead to higher biomass or grain weight. It has been argued that grain yield is poorly correlated with leaf photosynthetic rate when comparing between different genotypes [49]. However, the photosynthesis in ear parts showed significant contributions to wheat grain [11,50].

Chlorophyll content is considered as a key indicator of leaf greenness, and it is regularly used to explore leaf nutrient deficiency [51]. Lines with low chlorophyll content were always associated with insufficient regulation of linear electron transport and a limited ability to prevent over-reduction of PSI acceptor side regardless of the genotype, environment, and growth stage [52]. In this experiment, compared to Westonia, the chlorophyll content of the second leaves in Kauz was significantly higher before 14 DAA while it was slightly higher but not at a significant level in the flag leaves. The results of the flag leaf chlorophyll content were similar to our previous results in which the flag leaf chlorophyll was extracted in 85% acetone [6]. The chlorophyll of both leaves declined at 21 DAA. At 28 and 35 DAA, it was significantly lower in Kauz than in Westonia. In agreement with previous results, Kauz senesced earlier than Westonia [6]. It is well known that photosynthesis declines with leaf chlorophyll content according to plant age [53]. Regarding the decreasing chlorophyll content of the flag leaves over time, the significant lower levels of chlorophyll in Kauz resulted in considerably lower levels of glucose and fructose in the flag leaves in Kauz after 14 DAA relative to Westonia.

In agreement with previous research on Kauz and Westonia [9], a major component of the plant WSC is stored in the stem. In the current study, the WSC level in stems is double that in the leaves, and the accumulation of stem WSC is mainly based on the increment in fructan concentrations in the stems (82% of stem WSC). In the stems, before 14 DAA, the WSC and fructan concentrations in Kauz were significantly higher than those in Westonia. Furthermore, the levels of 1-kestose, 6-kestose and bifurcose were much higher in Kauz. A possible reason to account for this is that they result from the high levels of sucrose generated by photosynthesis in the second leaves. Relative to Westonia, the significantly higher levels of WSC in Kauz were not evident in field samples [7,9].

After 14 DAA, plants were experiencing WSC remobilization. Stem WSC and total fructan declined in Kauz, and after 21 DAA they declined in Westonia. The concentrations of 6-kestose and bifurcose dropped significantly in the stems of Kauz after 14 DAA, and this led to the significantly higher levels of fructose in the stems. Interestingly, 1-kestose concentrations were increased at 21 DAA. This indicates the remobilization of 2,6-linkage fructan is faster in Kauz under well-watered conditions, and implies different 6-FEH activities between Westonia and Kauz. Further investigation on 6-FEH function in stem WSC remobilization is required. Along with the degradation of fructan during grain filling, the grain weight in the main stems increased rapidly and the accumulation rate in Kauz was faster than that in Westonia between 21 to 28 DAA.

### 3.3. TaSUT1 Gene Expression Associated with the Diurnal Patterns of Sucrose Level

During grain filling, significant differences in leaf and stem sucrose levels were observed between Westonia and Kauz at 21 DAA—a critical stage of grain filling. Because of the direct relationship between sucrose level and sucrose transporters, sucrose transporter expression studies were recorded. In wheat, so far, the five SUT gene families have been studied, namely *TaSUT1*, *TaSUT2*, *TaSUT3*, *TaSUT4*, and *TaSUT5*. Compared with other SUT gene groups, *TaSUT1* is more highly expressed in all parts of the wheat plant [40]. High gene expression levels (1000 to 2000 units) for the three *TaSUT1* homeologous genes, namely, *TaSUT1_4A*, *TaSUT1_4B,* and *TaSUT1_4D,* were also detected in a RNAseq data set of a drought experiment [40]. Therefore, *TaSUT1* expression was focused using the same tissue samples for the sugar diurnal analysis at 21 DAA.

As observed for sugar diurnal patterns, there were clear diurnal patterns of *TaSUT1* expression. The correlation between the concentrations of sucrose and gene expression in leaves reveals that *TaSUT1* expression might be associated with sucrose levels in leaves. It is suggested that the significantly higher levels of *TaSUT1* gene expression together with the high levels of sucrose in Kauz would contribute to an increment in grain weight of Kauz between 21 to 28 DAA. Similar findings were reported in maize seedlings. The *ZmSUT1* expression level was positively associated with the concentrations of sucrose in 14-day-old maize leaves [36]. In maize, diurnal patterns of *ZmSUT2* were also clearly exhibited and the mutated zmsut2 plants accumulated two fold more sucrose, glucose, and fructose [38].

Recently, *ZmSWEET13* together with *ZmSUT1* were analyzed in maize and the photosynthesis of knock-out mutants was impaired, and high levels of soluble sugar accumulated in leaves [54]. These authors suggested the combined function of *ZmSWEET13* and *ZmSUT1* regulated sucrose transportation. It was argued that over-expression of sucrose transporters did not enhance grain yield under field conditions. In potato, over-expression of *SoSUT1* promoted sugar transport from leaves to tubers but did not result in higher yields [55]. In *Arabidopsis*, over-expression of *AtSUT2* enhanced sugar transport but plant growth was reduced [56]. On the contrary, using *AtSUT2* transformed rice, grain filling was accelerated in transgenic rice plants, and grain yield increased by 16% relative to wild-type plants in field trials [57]. A possible explanation of growth limitation is the shortage of a carbon source (sugar). It is suggested that enhanced yield relies on the combination of high sucrose transportation, enough carbon source and sink strength [56,58]. Relative to Westonia, the significantly higher gene expression of *TaSUT1* in Kauz was also detected in irrigated plants [40]. The higher levels of sucrose and *TaSUT1* expression may contribute to high grain yield in Kauz in well-watered conditions.

### 3.4. Sampling Time for WSC Analysis

Early research in our group showed that the total stem WSC levels were stable during the day after 10:00 but fluctuated in leaves (data not shown). Therefore, for investigating stem WSC remobilization, the stem samples were usually taken between 11:00 to 17:00 [59]. In this study, except for the stem WSC and fructan levels in Kauz at 14 DAA, there were no significant differences between 11:00 to 17:00 for stem WSC components in all developmental stages, but there were major differences in leaves. Also, the *TaSUT1* gene expression in Westonia stems showed significant differences between 12:30 and 16:30 at 21 DAA. Further investigation using field-grown plants is required for validating the results as the stem WSC levels and patterns during grain development differ between glasshouse and field [6,7].

## 4. Materials and Methods

### 4.1. Plant Materials

Wheat varieties Westonia and Kauz were used as described in a previous study [40]. Westonia, one of the top ten varieties sown in 2014 in Western Australia [60], is an early to mid-season maturity variety, [6], and produces high yield in medium and low rainfall areas. Kauz, generated in the International Maize and Wheat Improvement Center (CIMMYT, EI Batan, Mexico) [61], is a high yield variety in favourable environments and some drought conditions, and is thus considered as drought tolerant. However, under drought, the phenology showed that Westonia and Kauz had different mechanisms to achieve high yield, as Kauz visually senesced faster [6]. Kauz flowered 1–2 days later than Westonia and produced higher seed numbers per spike [6]. In the field, both varieties had high stem WSC concentration (40% of dry weight), whereas the remobilization patterns of stem WSC to grain differed [6,7].

### 4.2. Experimental Design

The experiment was carried out in a glasshouse at Murdoch University in Perth (32°04′ S; 115°50′ E), Australia. The glasshouse temperature was between 8–25 °C under natural light. The light was set at 500 µmol m^−2^s^−1^ and the glasshouse screens were computer controlled. The humidity was 60–70%. The light intensity, the day length, and the room temperature during sampling time are given in Appendix A. During the sampling time, the room temperature was between 11 to 22 °C. Based on the diurnal sampling time course (four hour (h) intervals, six times in 24 h, three biological replicates each time and seven developmental stages, and additional six pots for the final harvest), a total of 260 pots were prepared using two kg of potting mix with 1.3 g·kg^−1^ of Osmocote^®^ (Scotts Australia Pty Ltd., 24-32 Lexington Drive, Bella Vista New South Wales, Australia) and Grower’s Blue^®^ (Forbe Fertiliser Pty Ltd., 420 Nicholson Rd, Forrestdale, Western Australia, Australia) together with 0.7 g·kg^−1^ dolomite (CaMg(CO_3_)_2_) and 0.4 g·kg^−1^ of lime (Ca(OH)_2_). The potting mix comprised four parts of compost, two parts of pine bark, one-part coarse river sand, and one-part coco peat. Seeds of Westonia and Kauz were sown in autumn on 29 May 2017 and thinned to four plants per pot at the three leaf stage. Plants were well-watered until maturity via a capillary matting to the base of the containers. All pots were distributed randomly across benches.

### 4.3. Phenotyping and Sample Collection

Plants were sampled at heading when half of the plants within the variety were at this stage, then weekly from anthesis until 28 DAA. The samples harvested at the 7th developmental stage (35 DAA) had already senesced and were not used for WSC and gene expression measurements. On the sampling day, plants were collected six times at 08:30, 12:30, 16:30, 20:30, 00:30, and 04:30. Three replicates were harvested at each of the six timepoints. Each replicate consisted of four main tillers from four different plants (one pot). The flag leaf, the penultimate leaf (the second leaf), and main stems with heads were collected, placed in liquid nitrogen, then stored at −20 °C. Grain weight (GW), the total WSC and WSC components from the flag leaves (without sheaths), and main stems (all of the internodes above the soil surface with sheaths) were determined. The flag leaf length and width, plant height, tiller number, and plant biomass (dry) were recorded. The flag leaf area was calculated as leaf length × leaf width × 0.82 [62]. Three biological replicates were used for the final harvest at 45 DAA, grains were collected from main tillers of four plants in each replicate for measurement of GW, TGW, and KN per spike.

### 4.4. Chlorophyll Content Measurement

Leaf chlorophyll index was measured at heading and then weekly from anthesis until 35 DAA using a chlorophyll content meter (CCM-200) from Apogee instruments incorporation (www.apogeeinstruments.com). The mid area of the four flag leaves and four of the second leaves in each biological replicate were measured during the day between 12:30 to 14:30. The mean of the four leaf chlorophyll index formed one biological replicate and three mean values from three biological replicates were taken for statistical analysis.

### 4.5. Water Soluble Carbohydrate Analysis

WSCs were extracted from leaves and stem (sheath included) using boiling deionized water and quantified using anthrone reagent. The total WSC as % is expressed by weight of fructose per unit dry weight of the sample used in the extraction by dividing the dry weight of the sample extracted and then multiply with 100. The following formula was used to calculate the total WSC% [63]:% (Total soluble carbohydrates) = µg fructose (OD)/assay vol (mL) × extract volume (ml)/sample weight (g) × 10^−4^

The WSC components (glucose, fructose, sucrose, 1-kestose, 6-kestose, and bifurcose) of leaf and stem samples were measured using high-performance anion exchange chromatography with pulsed amperometric detection (HPAEC-PAD) after passing through a 0.3 mL bed volume of Dowex^®^−50 H^+^ and a 0.3 mL bed volume of Dowex^®^-1-acetate, followed by rinsing six times with 200 μL distilled water. Full details were described previously [7]. The total fructan concentration was calculated as the amount of the WSC concentration (as determined by the anthrone method) minus the concentration of glucose, fructose, and sucrose.

### 4.6. Gene Expression

Sub-samples taken from the same samples used for WSC and WSC component measurements were used for the gene expression. The RNA extraction, PCR amplification, and real time PCR were performed as described previously [40]. The total RNA was extracted by combining the protocol of TRIzol and RNA extraction kits, according to the instructions of the manufacturer (Invitrogen, Carlsbad, CA, and Qiagen, Victoria, Australia). RNA was quantified by UV spectrophotometry and confirmed by running on 1.5% agarose gel electrophoresis (Appendix A). DNA was removed from total RNA extracts by treatment with RNase free DNase I (Qiagen DNase Set) (Appendix A). A total of 200 ng of RNA for each sample was used for cDNA reverse transcription using High-Capacity cDNA Reverse Transcription Kits (Applied Biosystems) and random primers. Quantitative reverse transcription-PCRs (qRT-PCR) were performed in triplicate with the Power SYBR^®^ Green PCR Master Mix and contained 1 µl of 1:100 dilution of template cDNA in a Corbert Rotor-Gene RG-3000 (Corbett Research, Queensland, Australia). The *TaSUT1* gene primer pair used was based on the conserved region of *TaSUT1_4A*, *TaSUT1_4B,* and *TaSUT1_4D* but different from the gene sequences of other four *TaSUT* gene groups (forward sequence: TGGATTCTGGCTCCTTGAC and reverse sequence: GCCATCCAAGAACAGAAGATT) [40]. The PCR conditions were one cycle at 95 °C for 10 min, 40 cycles at 92 °C for 15 s, and 58 °C for 60 s. Three biologic replicates were measured at each timepoint and three technical replicates of each biologic replicate were performed for the gene expression. The relative quantification-standard curve method was used for the gene expression calculation (http://www.thermofisher.com/au/en/home/life-science/pcr/real-time-pcr-learning-center). The amplification of the cytosolic glyceraldehyde-3-phosphate dehydrogenase (*GAPDH*) of the same sample was used as an internal reference to standardize the amount of sample DNA added to a reaction. Different dilutions of the *TaSUT1* gene plasmid were used for the gene expression relative standard curve for both the target and the internal reference. For each experimental sample, the amount of target and internal reference is determined from the appropriate standard curve, then the target amount is divided by the internal reference amount to obtain a normalized target value. The *TaSUT1* gene expression units per sample (one biologic replicate) were the average level of the three technical replicates divided by the average level of the *GAPDH* expression level in three technical triplicates. *TaSUT1* gene expression in the stems and flag leaves were measured in three biologic replicates.

### 4.7. Statistical Analysis

Statistical analysis was conducted using IBM SPSS (ANOVA, Australian Head Office IBM Australia Ltd., IBM Centre, 601 Pacific Highway, St Leonards New South Wales, Australia). A post-hoc Tukey’s test was used for analysis of group (three biological replicates) significance. The student’s *t*-test was used for pair analysis of the data. Correlation analysis was determined by Pearson bivariate in IBM SPSS. The daily average concentrations of WSC and WSC components in the flag leaves and stems were used for presenting the patterns during developmental stages. Graphs were generated using sigma plot 13.0.

## 5. Conclusions

Diurnal effects on WSC and WSC components in leaves and stems of Westonia and Kauz were quantified from heading until 28 DAA in a glasshouse experiment. Diurnal patterns in WSC, sucrose, glucose and fructose in leaves were observed but not for 6-kestose and bifurcose. In stems, diurnal patterns were present for WSC and fructan levels in Kauz at 14 DAA. Significant correlations between levels of *TaSUT1* expression and sucrose in leaves at 21 DAA suggest that *TaSUT1* expression is influenced by the size of leaf sucrose pool. High levels of *TaSUT1* expression and sucrose in Kauz may contribute to its high grain yield under well-watered conditions. Further investigation using field materials is required for validating the diurnal results.

## Figures and Tables

**Figure 1 ijms-21-08276-f001:**
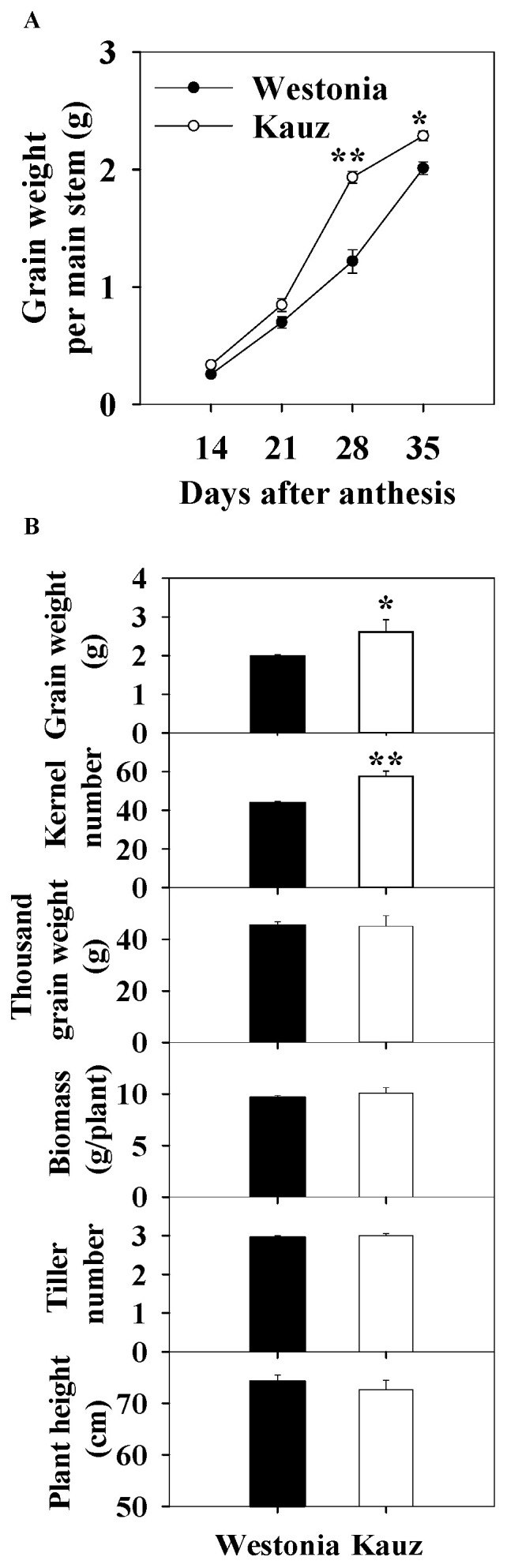
Grain accumulation (**A**) and kernel number per spike, grain weight per spike, thousand grain weight), biomass per plant (dry weight), tiller number per plant, and plant height of Westonia and Kauz at the final harvest (grain maturity) (**B**). The vertical bars represent standard errors. Asterisks (*) and (**) represent significant levels at *p* < 0.05 and *p* < 0.01, respectively.

**Figure 2 ijms-21-08276-f002:**
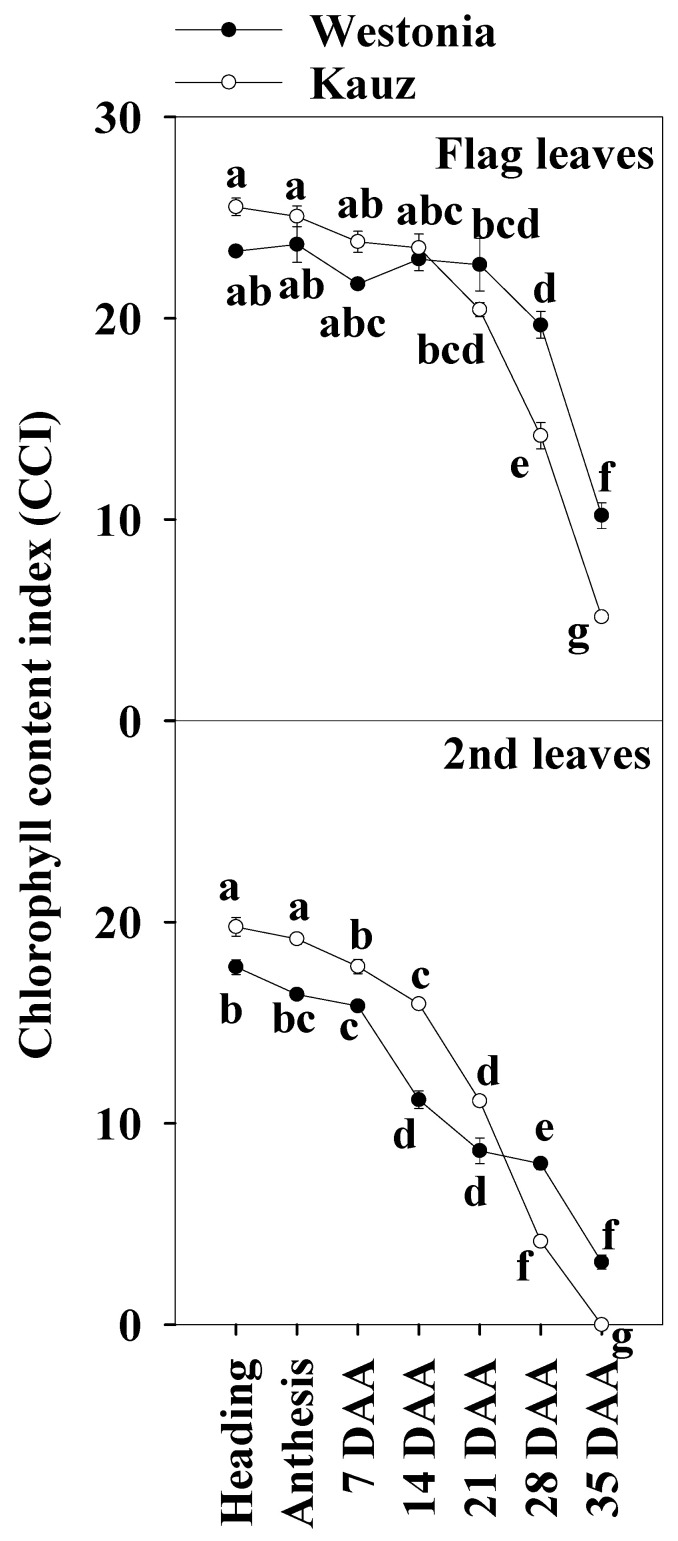
Chlorophyll content index of the flag and second leaves in Westonia and Kauz across grain developmental stages. The vertical bars represent standard errors. Values with the same letter are statistically not different at *p* = 0.05.

**Figure 3 ijms-21-08276-f003:**
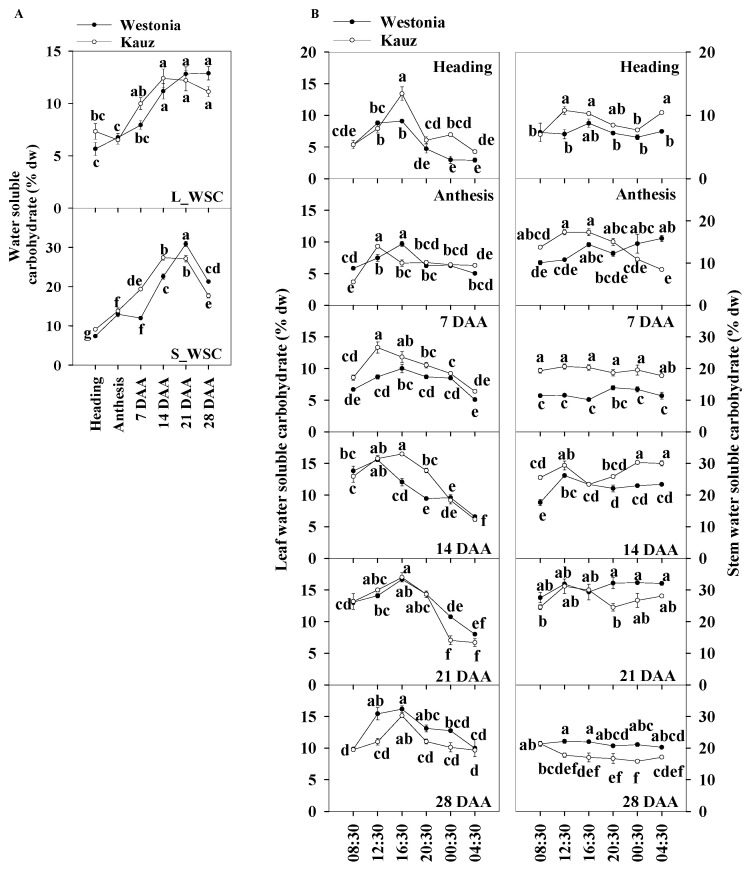
Water soluble carbohydrate concentrations and diurnal changes in the flag leaves and main stems across grain developmental stages in Westonia and Kauz. (**A**) Water soluble carbohydrate concentrations in the flag leaves (L_WSC) and main stems (S_WSC); (**B**) Water soluble carbohydrate diurnal analysis in flag leaves (**left**) and main stems (**right**). The vertical bars represent standard errors. Values with the same letter are statistically not different at *p* = 0.05.

**Figure 4 ijms-21-08276-f004:**
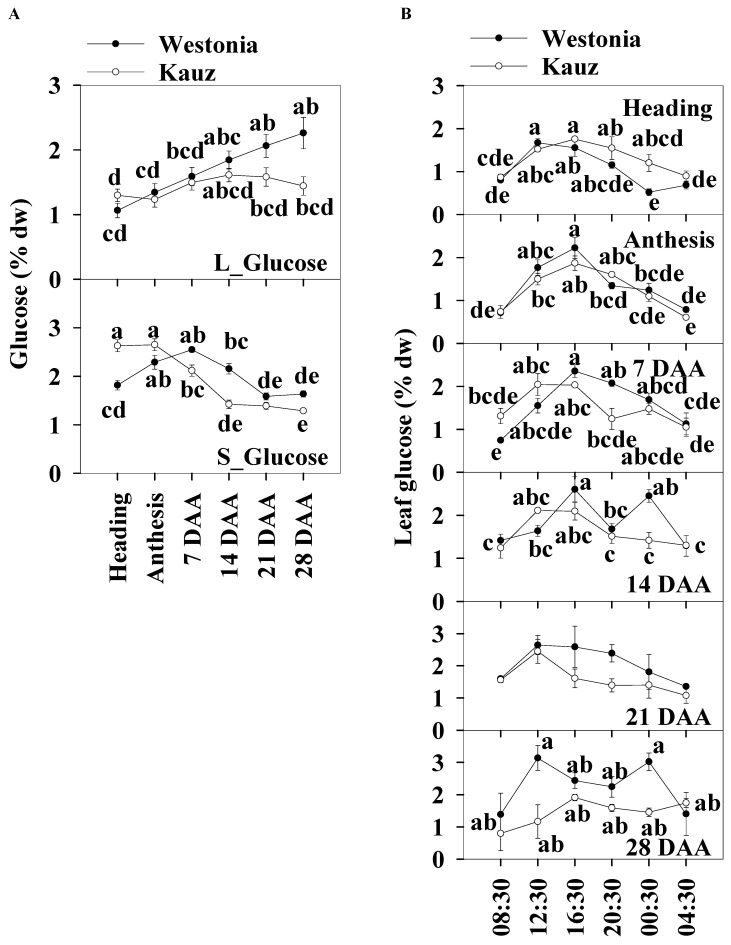
Glucose concentrations and diurnal changes in flag leaves and main stems across grain developmental stages in Westonia and Kauz. (**A**) Glucose concentrations in the flag leaves (L_Glucose) and main stems (S_Glucose); (**B**) Glucose diurnal analysis in the flag leaves. The vertical bars represent standard errors. Values with the same letter are statistically not different at *p* = 0.05.

**Figure 5 ijms-21-08276-f005:**
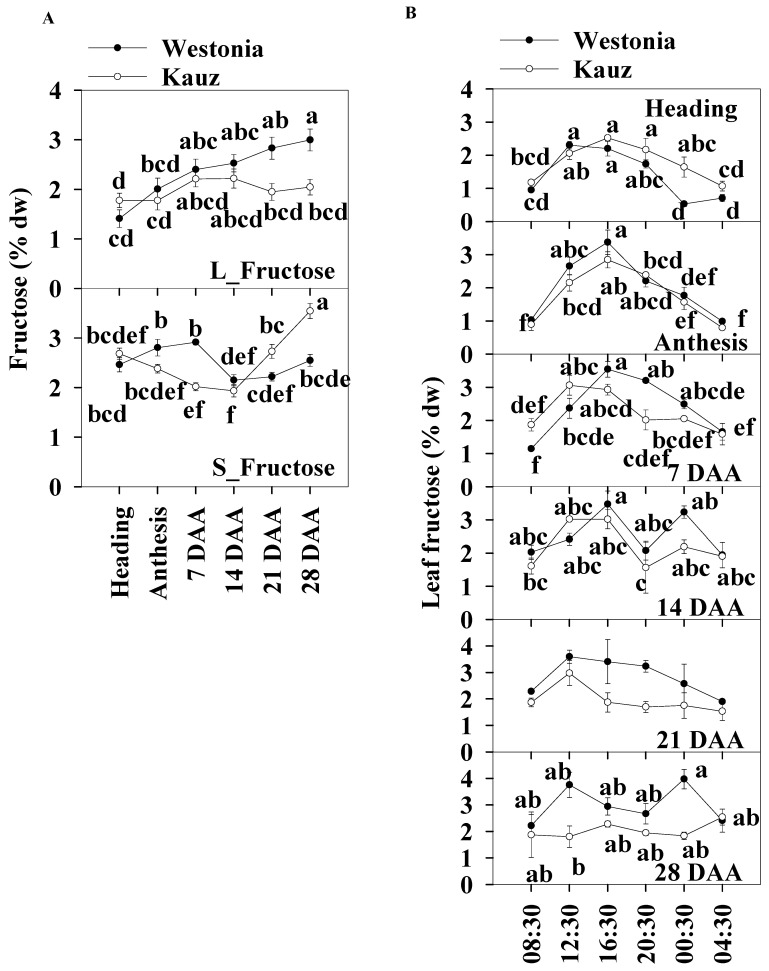
Fructose concentrations and diurnal changes in flag leaves and main stems across grain developmental stages in Westonia and Kauz. (**A**) Fructose concentrations in the flag leaves (L_Fructose) and main stems (S_Fructose); (**B**) Fructose diurnal analysis in the flag leaves. The vertical bars represent standard errors. Values with the same letter are statistically not different at *p* = 0.05.

**Figure 6 ijms-21-08276-f006:**
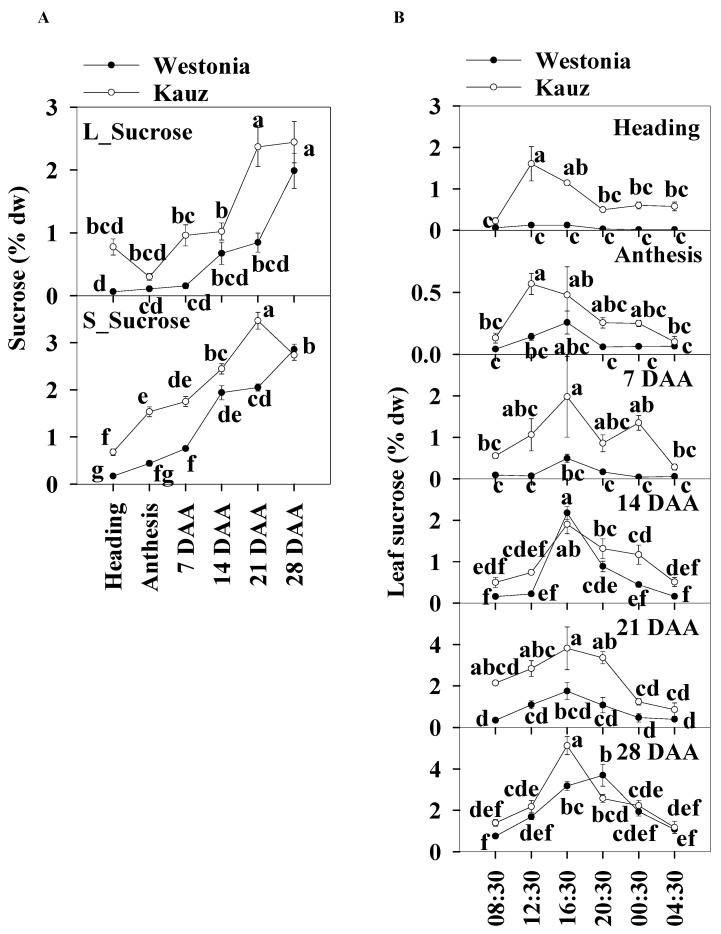
Sucrose concentrations and diurnal changes in flag leaves and main stems across grain developmental stages in Westonia and Kauz. (**A**) Sucrose concentrations in the flag leaves (L_Sucrose) and main stems (S_Sucrose); (**B**) Sucrose diurnal analysis in the flag leaves. The vertical bars represent standard errors. Values with the same letter are statistically not different at *p* = 0.05.

**Figure 7 ijms-21-08276-f007:**
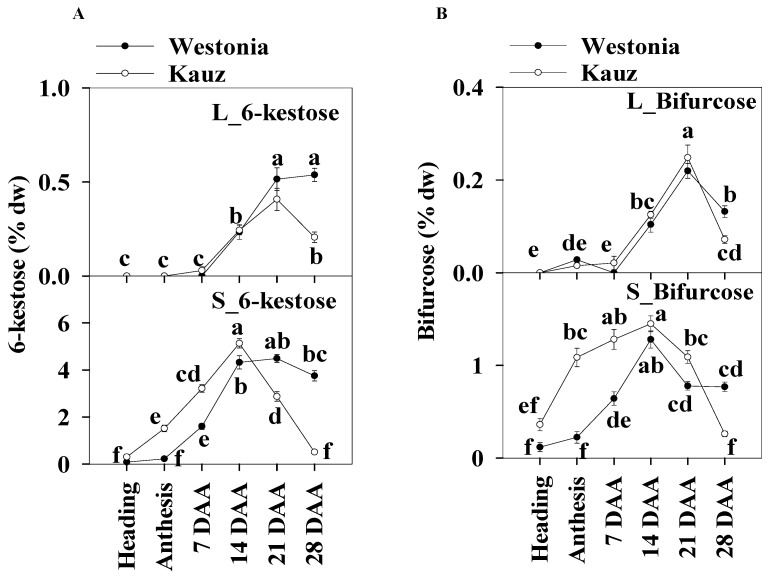
6-kestose and bifurcose concentrations in flag leaves and main stems across grain developmental stages in Westonia and Kauz. (**A**) 6-kestose concentrations in the flag leaves (L_6-kestose) and main stems (S_6-kestose); (**B**) Bifurcose concentrations in the flag leaves (L_bifurcose) and main stems (S_bifurcose). The vertical bars represent standard errors. Values with the same letter are statistically not different at *p* = 0.05.

**Figure 8 ijms-21-08276-f008:**
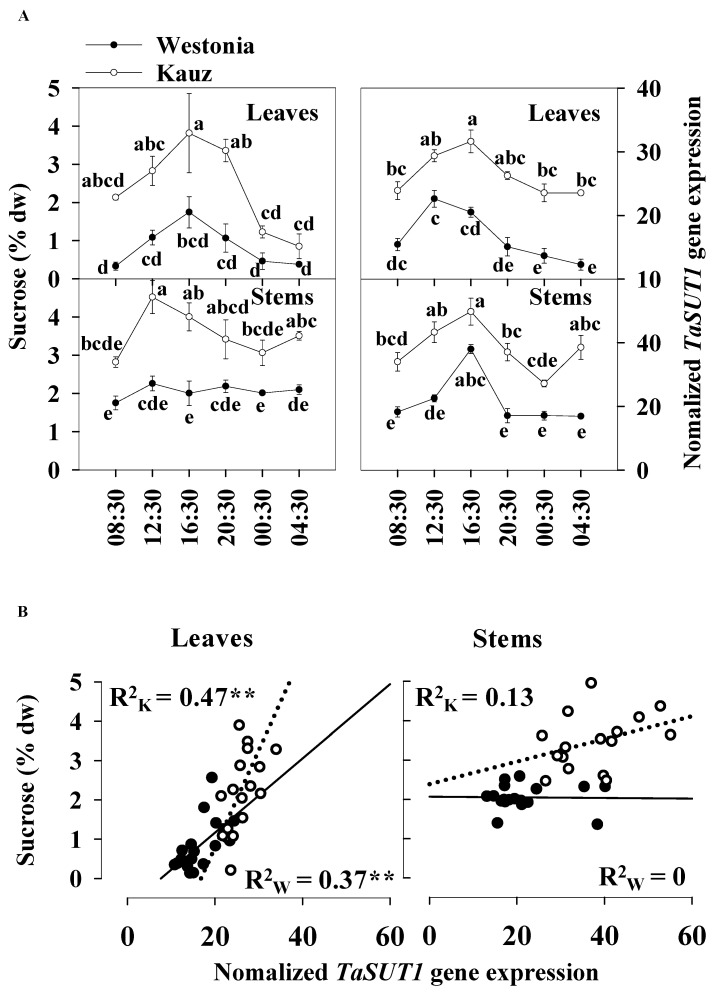
Associations between the diurnal levels of sucrose and *TaSUT1* gene expression of leaves and stems at 21 DAA in Westonia and Kauz. (**A**) Sucrose diurnal analysis in flag leaves and main stems and diurnal analysis of *TaSUT1* gene expression in flag leaves and main stems; (**B**) The correlations between the levels of sucrose and *TaSUT1* gene expression in flag leaves and main stems in Westonia (R^2^_w_) and Kauz (R^2^_k_), respectively. Three biological replicates were measured at each timepoint and three technical replicates of each biological replicate were performed for the gene expression. The vertical bars represent standard errors. Values with the same letter are statistically not different at *p* = 0.05. Asterisks (**) represent the significant levels at *p* < 0.01.

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
