# Peer review of "Diurnal Changes in Water Soluble Carbohydrate Components in Leaves and Sucrose Associated TaSUT1 Gene Expression during Grain Development in Wheat"

_ijms, 2020, doi:10.3390/ijms21218276_

Round 1

Reviewer 1 Report

In the article by Sarah Al-Sheikh Ahmed et al., the quantitative analysis of soluble sugars in leaves and stems of two wheat cultivars differing in productivity is presented. The topic is of practical importance since the questions related to the productivity of one of the major crop cultures are studied. The article content is relevant to the Special Issue topic.

Although the study is mostly descriptive, the presented detailed quantitative analysis of water-soluble sugars can provide a good basis for the further identification of the regulatory mechanisms responsible for wheat productivity. High level of TaSUT1 expression and an increased sucrose content are assumed to be the factors contributing to high grain yield in Kauz.

The manuscript is well written.

Minor comments:

Chlorophyll content was taken as a measure of photosynthetic activity. It is a rough approach. The analysis of photosynthetic activity through the other available simple methods would significantly improve the value of the generated data.

Additional information about day length during the collection of samples would be helpful for the data interpretation.

Light intensity is not taken into account, but this factor can influence WSC content, especially in leaves.  

Line 43 The regulatory mechanisms underlying the inhibition of photosynthesis by sucrose are not described accurately: “Sucrose accumulation in leaves inhibits photosynthesis since the inorganic phosphate molecules released from sucrose synthesis are used for the synthesis of more triose phosphate in chloroplasts.”

Line 69 “whether correlations sucrose levels are correlated with TaSUT1 gene expression”. Please, remove the word “correlations”

Line 314 “Compared with other SUT gene families” Should it be “gene family members”?

Author Response

Review 1

Reviewer: The manuscript is well written.

Minor comments:

Chlorophyll content was taken as a measure of photosynthetic activity. It is a rough approach. The analysis of photosynthetic activity through the other available simple methods would significantly improve the value of the generated data.

Response: We agree to the reviewer’s comments. The flag leaf chlorophyll content data measured in this study was similar to our previous results where the flag leaf chlorophyll was extracted in 85% acetone. The following sentence was added to the text (line 303-305)

Line 303-305: The results of the flag leaf chlorophyll content were similar to our previous results in which the flag leaf chlorophyll was extracted in 85% of acetone (Zhang et al. 2009)

Reviewer: Additional information about day length during the collection of samples would be helpful for the data interpretation.

Response: The day length figure during sampling time was added (Supplementary figure S9). The following text was added (line 388-391).

“The light intensity and the day length during sampling time are given in supplementary figure S9.”

Reviewer: Light intensity is not taken into account, but this factor can influence WSC content, especially in leaves.  

Response: The light intensity is important, and the light was set at 500 µmol m-2s-1. This was achieved by computer-controlled screens in the glasshouse roof (text was added in line 388-391). The humidity was 60-70%. The light intensity during the sampling time was added. See the response above. The text was added (line 388-391).

Reviewer: Line 43 The regulatory mechanisms underlying the inhibition of photosynthesis by sucrose are not described accurately: “Sucrose accumulation in leaves inhibits photosynthesis since the inorganic phosphate molecules released from sucrose synthesis are used for the synthesis of more triose phosphate in chloroplasts.”

Response:  We have changed the text to (line 57-67):

Line 57-67: Sucrose is synthesized in photosynthetic leaves from the triose phosphate produced in the Calvin cycle, with another triose phosphate - glyceraldehyde 3-phosphate to form fructose 1,6-bisphosphate, which is converted to fructose 6-phosphate by cytosolic fructose 1,6-bisphosphatase, then synthesized to sucrose with enzymes of the glucose 6-phosphate isomerase (phosphoglucose isomerase), phosphoglucomutase, uridine diphosphate‐glucose pyrophosphorylase, sucrose phosphate synthase (SPS) and sucrose phosphate phosphatase (Halford et al. 2011). The activity of SPS increases in parallel with photosynthesis rate. When exogenous sucrose is supplied, SPS activity decreases. Therefore, too much sucrose accumulated in leaves inhibits photosynthesis (Stitt et al. 1988; Halford et al. 2011).

Reviewer: Line 69 “whether correlations sucrose levels are correlated with TaSUT1 gene expression”. Please, remove the word “correlations”

Response: The “correlations” was deleted (line 103).

Reviewer: Line 314 “Compared with other SUT gene families” Should it be “gene family members”?

Response: The text was changed to “other SUT gene groups” (line 333).

Reviewer 2 Report

I believe the presentation of the paper is not sufficiently advanced to warrant publication. Personally, with this dataset, I don't think the authors have any chance to improve the value of their manuscript.
Sincerely, related to the present form I have found principally some weakness. Paper is very pragmatic and conventional. Clearly, considerable effort has been made with the manuscript. Paper doesn't offer a new hypothesis. Authors characterized only the differences of 2 wheat genotypes in the dynamics of water-soluble carbohydrate remobilization to the grain. This problem is not new. They didn't explain enough, why they selected just these varieties.
The manuscript is the only collection of a few parameters. The creativity of authors and the new innovative view is missing. The understanding of mechanisms is very limited, as it is restricted to papers that have a particular view and deliberately ignore alternatives, and does not present a balanced view of the evidence.
My major critical remarks refer to the complexity of the paper. Many inappropriate parts of the paper exist.
These encompass:
- lack of hypothesis
- authors didn't characterize well biology and physiology of the genotypes
- authors do diurnal analysis without environmental analysis and also without some key physiological measurements describing ontogenetic changes of leaves - plants
- missing relevant references
I have found some major flaws:
- the topic is not new at all.
-only very basic parameters were determined. In the era of omic sciences, I would expect this manuscript addressed to IJMS methodologically better and deeper paper.

Author Response

Review 2

Reviewer: My major critical remarks refer to the complexity of the paper. Many inappropriate parts of the paper exist.
These encompass:
- lack of hypothesis

Response: The hypothesis is in line 103-105: “The overall hypothesis is that sucrose transporter expression and tissue concentrations of WSC are affected by the biological clock and the stages of grain development.” The hypothesis as the second sentence is also added to the abstract.

Reviewer: - authors didn't characterize well biology and physiology of the genotypes

Response: Information about the biology and physiology of the genotypes was added in the manuscript: (line 374-385):

Line 374-385: Wheat varieties Westonia and Kauz were used as described in a previous study (Al-Sheikh Ahmed et al. 2018). Westonia, one of the top ten varieties sown in 2014 in Western Australia (Trainor et al. 2015), is an early to mid-season maturity variety, (Zhang et al. 2009), and produces high yield in medium and low rainfall areas. Kauz, generated in the International Maize and Wheat Improvement Center (CIMMYT, EI Batan, Mexico) (Butler et al. 2005), is a high yield variety in favourable environments and some drought conditions, thus considered as drought tolerance. However, under drought, the phenology showed that Westonia and Kauz had different mechanisms to achieve high yield as Kauz visually senesced faster (Zhang et al. 2009). Kauz flowered 1-2 days later than Westonia and produced higher seed number per spike (Zhang et al. 2009). In the field, both varieties had high stem WSC concentration (40% of dry weight), whereas the remobilization patterns of stem WSC to grain  were different (Zhang et al. 2009; Zhang et al. 2015).

Reviewer: - authors do diurnal analysis without environmental analysis and also without some key physiological measurements describing ontogenetic changes of leaves – plants

Response: Analysis based on the environmental factors of day length, light intensity, the time of sunrise and sunset have been added (Supplementary figure S9). The flag leaf area of Westonia and Kauz and its diurnal data were added (Supplementary figure S1). Text was added into result (line 118-124), discussion sections (Line 287-297) and the section of Materials and methods (line 388-391; line 407-409).

Line 118-124: The average leaf area of the flag leaves sampled daily were used for presenting the flag leaf area patterns across developmental stages (Supplementary figure S1A). The flag leaf area grandually expanded from heading until 7 and 14 DAA in Kauz and Westonia, respectively. After a drop at 21 DAA, the values of the flag leaf area returned to previous levels at 28 DAA in both varieties. Overall, the flag leaf area of Westonia was significantly higher than in Kauz (Supplementary figure S1A). No diurnal changes were identified in the flag leaf area of either variety (Supplementary figure S1B).

Line 289-299: Generally, there is a positive correlation between photosynthetic capacity (Am) and specific leaf area (Gulias et al. 2003). Later, researchers identified that the relationship of leaf area and plant biomass was not linear but varied depending on carbon partitioning (Weraduwage et al. 2015). In addition, leaf thickness is a factor in carbon partitioning as thicker leaves have greater capacity to fix carbon than thin leaves (Chabot et al. 1979; Lambers et al. 2008). Although the flag leaf area of Westonia was significantly higher than in Kauz, this may not necessarily indicate a higher photosynthetic capacity in Westonia. Our previous glasshouse study showed that the rate of photosynthesis was slightly lower in Westonia than in Kauz (Zhang et al. 2009). We did not measure leaf thickness, so it is not known whether Kauz leaves are thicker than Westonia. Moreover, the 2nd leaf is important for carbon export, but we did not measure the area of these leaves. The current results showed that the higher flag leaf area in Westonia did not lead to higher biomass or grain weight.  

Line 388-391: The light was set at 500 µmol m-2s-1 and the glasshouse screens were computer controlled. The humidity was 60-70%. The light intensity and the day length during sampling time are given in supplementary figure S9.

Line 410-412: The flag leaf length and width, plant height, tiller number and plant biomass (dry) were recorded. The flag leaf area was calculated as leaf length × leaf width × 0.82 (Chen et al. 2019b)

Reviewer: - missing relevant references

Response: Thank you. Nine more references were added. Please see the following:

Chen T, Li G, Islam MR, Fu W, Feng B, Tao L, Fu G (2019) Abscisic acid synergizes with sucrose to enhance grain yield and quality of rice by improving the source-sink relationship BMC Plant Biol 19:525

Stitt M, Wilke I, Feil R, Heldt HW (1988) Coarse control of sucrose-phosphate synthase in leaves: Alterations of the kinetic properties in response to the rate of photosynthesis and the accumulation of sucrose Planta 174:217-230

Rezaul IM, Baohua F, Tingting C, Weimeng F, Caixia Z, Longxing T, Guanfu F (2019) Abscisic acid prevents pollen abortion under high-temperature stress by mediating sugar metabolism in rice spikelets Physiol Plantarum 165:644-663

Wang L, Lu Q, Wen X, Lu C (2015) Enhanced Sucrose Loading Improves Rice Yield by Increasing Grain Size Plant Physiol 169:2848-2862

Zhang C-X et al. (2018) Heat stress-reduced kernel weight in rice at anthesis is associated with impaired source-sink relationship and sugars allocation Environ Exp Bot 155:718-733

Gulias, J.; Flexas, J.; Mus, M.; Cifre, J.; Lefi, E.; Medrano, H. Relationship between maximum leaf photosynthesis, nitrogen content and specific leaf area in balearic endemic and non-endemic mediterranean species. Ann Bot 2003, 92, 215-222, doi:10.1093/aob/mcg123.

Weraduwage, S.M.; Chen, J.; Anozie, F.C.; Morales, A.; Weise, S.E.; Sharkey, T.D. The relationship between leaf area growth and biomass accumulation in Arabidopsis thaliana. Front. Plant Sci. 2015, 6, doi:10.3389/fpls.2015.00167.

Chabot, B.F.; Jurik, T.W.; Chabot, J.F. Influence of instantaneous and integrated light-flux density on leaf anatomy and photosynthesis. Am. J. Bot. 1979, 66, 940-945, doi:10.1002/j.1537-2197.1979.tb06304.x.

Lambers, H.; Chapin, F.S.; Pons, T.L. Plant physiological ecology; Springer: New York, NY, 2008.

Reviewer: I have found some major flaws:
- the topic is not new at all.

Response: We agree with the reviewer’s comments in that the overall topic has been explored in some plants. However, the diurnal changes of WSC components and SUT1 gene expression are new in wheat and we believe this contribution is useful.

Our results revealed clear diurnal patterns of WSC and sucrose levels in leaves but not in stem. Compared with the leaf WSC, the significantly high levels of stem WSC confirmed the WSC storage function in stem. High correlations between levels of TaSUT1 expression and sucrose in leaves were observed. Significantly genotypic differences in fructan accumulation and degradation indicate the differentiated function levels of enzymes in fructan synthase and degrade pathways. This greatly advances our current understanding of the role of WSC remobilization in grain filling.

Reviewer: -only very basic parameters were determined. In the era of omic sciences, I would expect this manuscript addressed to IJMS methodologically better and deeper paper.

Response: It is true in that basic parameters have been measured and recorded, but a massive work was involved in this manuscript. The diurnal results in wheat were new and robust. Following reviewer’s suggestions, we added more parameters and references in the manuscript. The introduction, results and discussions were revised accordingly. It is hoped that in the future, research funds will become available for a more in depth omic approach to pre-breeding of drought tolerant bread wheat.

Reviewer 3 Report

This manuscript described fluctuations of water-soluble carbohydrate (WSC) and a sucrose transporter (SUT) gene expression during developmental stages from heading to maturity in two wheat varieties of Westonia and Kauz. The authors collected frag leaves and main stems in the two varieties and evaluated contents of WSC such as fructan, glucose, sucrose, fructose, 1-kestose, 6-kestose and bifurcose. The levels of the WSC were associated with grain weight and chlorophyll content. Especially, they observed high correlation between sucrose content and mRNA of TaSUT1 in frag leave in Kauz, which showed increasing chlorophyll content at early grain filling stages and final grain weight. This study indicated importance of diurnal pattern of WSCs and gene expressions. However, the authors should add and revise several descriptions in this manuscript described below.

1) A lot of previous studies reported functional characterization of SUT genes in various plant species. For examples, Wang et al. (2015) indicated that expression of AtSUC2 gene in rice increased sucrose phloem loading, grain filling rate and final grain size. And, as well as grain weight traits, it is also important to evaluate plant biomass including plant height and tiller number, because chlorophyll content and leaf senescence associated with WSC transportation and recycling. The authors also should investigate these phenotypes in the two wheat varieties Westonia and Kauz in this study.

Liang Wang, Qingtao Lu, Xiaogang Wen, Congming Lu
Enhanced sucrose loading improves rice yield by increasing grain size.
Plant Physiol (2015) 169(4):2848-2862.

2) As the authors mentioned, expression levels of SUC genes affect to other gene expressions involved in starch biosynthesis, starch degradation and sucrose transport such as SWEET genes. Therefore, the authors should investigate mRNA expressions of these other genes. qPCR and RNA-seq techniques would be useful for comprehensive transcriptome analysis. Actually, in this study, mRNA expression of TaSUT1 didn't show significant correlation with sucrose content. These results might indicate existence of other important associations between genes and WSCs content in this study. The authors should indicate mRNA expressions of other SUC family genes and SWEET genes at least.

3) Chen et al. (2019) reported ABA was also one of molecular signals in response to abiotic stress such as drought, as well as sucrose. Comprehensive transcriptome analysis could also reveal relationships between the ABA-related gene and abiotic stress conditions.

Tingting Chen, Guangyan Li, Mohammad Rezaul Islam, Weimeng Fu, Baohua Feng, Longxing Tao, Guanfu Fu
Abscisic acid synergizes with sucrose to enhance grain yield and quality of rice by improving the source-sink relationship.
BMC Plant Biol (2019) 19(1):525.

4) The authors should describe correlation relationships among all of WSC components, agronomic traits and mRNA expressions of the TaSUT1 genes and other important genes. The authors only investigated correlations between expression of the TaSUT1 gene and sucrose content in Figure8B in this manuscript. However, a lot of factors would be involved in the sucrose transportation from aged leaf and stem to embryo in individual plant. Therefore, statistical analysis considering the whole evaluated factors (traits) would be effective to consider their correlations in detail.

5) L. 234. Full name of the abbreviation 'SUT' should indicate the first position of this manuscript.

6) L. 375-L. 377. If the authors investigated plant height as described here, you should indicate these results in Figure 1.

7) L. 398-L. 399. Wheat has three genomes and three homeologous genes of TaSUT1. The previous studies of Al-Sheikh Ahmed et al. (2018) separately investigated mRNA expressions of the three homeologous TaSUT1 genes. Why and how did you integrate their expressions as one TaSUT1 gene in this study?

Sarah Al-Sheikh Ahmed, Jingjuan Zhang, Wujun Ma, Bernard Dell
Contributions of TaSUTs to grain weight in wheat under drought.
Plant Mol Biol (2018) 98(4-5):333-347.

Author Response

Review 3

Reviewer: 1) A lot of previous studies reported functional characterization of SUT genes in various plant species. For examples, Wang et al. (2015) indicated that expression of AtSUC2 gene in rice increased sucrose phloem loading, grain filling rate and final grain size. And, as well as grain weight traits, it is also important to evaluate plant biomass including plant height and tiller number, because chlorophyll content and leaf senescence associated with WSC transportation and recycling. The authors also should investigate these phenotypes in the two wheat varieties Westonia and Kauz in this study.

Liang Wang, Qingtao Lu, Xiaogang Wen, Congming Lu
Enhanced sucrose loading improves rice yield by increasing grain size.
Plant Physiol (2015) 169(4):2848-2862.

Response: Thanks, the reference of Wang et al. has been added (line 354-355). The plant biomass including plant height and tiller number has been included in Figure 1 and the text was changed in line 410-412.

Line 354-355: Using AtSUT2 transformed rice, grain filling was accelerated in transgenic rice plants and grain yield increased by 16% relative to wild-type plants in field trials (Wang et al. 2015).

Line 410-412: The plant height, tiller number and plant biomass (dry) were recorded.

Reviewer: 2) As the authors mentioned, expression levels of SUC genes affect to other gene expressions involved in starch biosynthesis, starch degradation and sucrose transport such as SWEET genes. Therefore, the authors should investigate mRNA expressions of these other genes. qPCR and RNA-seq techniques would be useful for comprehensive transcriptome analysis. Actually, in this study, mRNA expression of TaSUT1 didn't show significant correlation with sucrose content. These results might indicate existence of other important associations between genes and WSCs content in this study. The authors should indicate mRNA expressions of other SUC family genes and SWEET genes at least.

Response: We agree with reviewer’s comments. Many genes in WSC remobilization and grain filling pathway need to be analysed. Because of the limited time for the first author – Sarah Al-Sheikh Ahmed – a PhD student, she focused on TaSUT1 gene expression for this study as TaSUT1 was the major gene within the five TaSUT groups for sucrose remobilization. In her study, she identified the significantly positive correlations between TaSUT1 gene expression and leaves (line 261), but not the stem. In wheat stem, there may be other sucrose transporting mechanisms involved, such as SWEET genes.

Reviewer: 3) Chen et al. (2019) reported ABA was also one of molecular signals in response to abiotic stress such as drought, as well as sucrose. Comprehensive transcriptome analysis could also reveal relationships between the ABA-related gene and abiotic stress conditions.

Tingting Chen, Guangyan Li, Mohammad Rezaul Islam, Weimeng Fu, Baohua Feng, Longxing Tao, Guanfu Fu
Abscisic acid synergizes with sucrose to enhance grain yield and quality of rice by improving the source-sink relationship.
BMC Plant Biol (2019) 19(1):525.

Response: Thanks, the reference was added in line 67-75.

Line 67-75: Sucrose is a signalling molecule in mediating source-sink relationship as exogenous sucrose reduced the impact of heat-stress on spikelet fertility, kernel weight, and plant dry weight accumulation at flowering in rice (Zhang et al. 2018). The role of sucrose as a signalling molecule in plants may involve interactions with endogenous plant hormones, such as ABA as it was reported that ABA could enhance sugar metabolism and transport into spikelets of rice to prevent pollen abortion by increasing the expression level of SUT1, INV and SUS genes at the meiosis stage in pollen mother cells (Rezaul et al. 2019). Exogenous sucrose and ABA could significantly enhance the starch contents and the enzyme activities involved in starch synthesis in grains (Chen et al. 2019a).

Reviewer: 4) The authors should describe correlation relationships among all of WSC components, agronomic traits and mRNA expressions of the TaSUT1 genes and other important genes. The authors only investigated correlations between expression of the TaSUT1 gene and sucrose content in Figure8B in this manuscript. However, a lot of factors would be involved in the sucrose transportation from aged leaf and stem to embryo in individual plant. Therefore, statistical analysis considering the whole evaluated factors (traits) would be effective to consider their correlations in detail.

Response: We agree with reviewer’s comments. Sucrose remobilization in plants is an important aspect for all agronomic traits. We think the relationship between sucrose level and sucrose transporters are the closest and direct relationship. This is the focused topic in this study.

Reviewer: 5) L. 234. Full name of the abbreviation 'SUT' should indicate the first position of this manuscript.

Response: The abbreviation of ‘SUT’ has been indicated in abstract (line 24).

Reviewer: 6) L. 375-L. 377. If the authors investigated plant height as described here, you should indicate these results in Figure 1.

Response: Plant biomass, tiller number and plant height have been added to Figure 1. The following text was added in the results (line 114-115) and materials and methods (line 410-412)

Line 114-115: The differences in biomass, tiller number and plant height were not significant.

Line 410-412: The flag leaf length and width, plant height, tiller number and plant biomass (dry) were recorded. The flag leaf area was calculated as leaf length × leaf width × 0.82.

Reviewer: 7) L. 398-L. 399. Wheat has three genomes and three homeologous genes of TaSUT1. The previous studies of Al-Sheikh Ahmed et al. (2018) separately investigated mRNA expressions of the three homeologous TaSUT1 genes. Why and how did you integrate their expressions as one TaSUT1 gene in this study?

Sarah Al-Sheikh Ahmed, Jingjuan Zhang, Wujun Ma, Bernard Dell
Contributions of TaSUTs to grain weight in wheat under drought.
Plant Mol Biol (2018) 98(4-5):333-347.

Response: The gene expression levels of the three homeologous TaSUT1_4A, TaSUT1_4B and TaSUT1_4D in our previous paper (Al-Sheikh Ahmed et al. 2018) was from a RNAseq data set of a drought experiment. In this study, we would like to see whether there were diurnal patters in TaSUT1 gene expression and the correlations between sucrose levels and the TaSUT1 gene expression levels. We used the primers based on the conserved region of TaSUT1_4A, TaSUT1_4B and TaSUT1_4D but different from the gene sequences of other four TaSUT gene groups. Please see the primer sequences and the primer regions in Table 1 and Fig. 2 in Al-Sheikh Ahmed et al. 2018. The following sentence was added in Materials and Methods section (line 439-441).

Line 439-441: TaSUT1 gene primer pair (forward sequence: TGGATTCTGGCTCCTTGAC and reverse sequence: GCCATCCAAGAACAGAAGATT) (Al-Sheikh Ahmed et al. 2018) was used.

Round 2

Reviewer 2 Report

I do not agree with sentences: "The chlorophyll content of the flag leaf is considered to be a key determinant of photosynthetic capacity [11]. Furthermore, high chlorophyll content is related to high grain yield".
Authors probably don't understand well the mechanisms of photosynthesis. The net assimilation rate is not directly correlated with pigment content.
I recommend significantly improve the introduction and discussion section focussed on the primary processes of biomass production.
Read/ use the papers of J.Araus, M.Zivcak, M.Brestic, E.Murchie, S.Landjeva, Dobrikova, P.Petrov, A. Borner, etc.
The authors compared only 2 genotypes of wheat. This research is not useful for the next applied research and breeding.
The current results showed that the higher flag leaf area in Westonia did not lead to higher biomass or grain weight. Many authors published correlations that higher leaf area of flag leaf is one of the most important factors correlated with grain weight. The authors must improve the discussion and include new references.

Author Response

Response:

1)     Following reviewer’s suggestions, the following sentences were deleted. "The chlorophyll content of the flag leaf is considered to be a key determinant of photosynthetic capacity [11]. Furthermore, high chlorophyll content is related to high grain yield".

2)     In the introduction (line 46-52) and discussion (line 336-344; line 347-349), the following additional texts were added by following the reviewer’s suggestions.

3)     Line 46-52: Photosynthesis from leaves, sheath, culm and ears contributes to the total photosynthetic capacity in wheat (Sanchez-Bragado et al., 2016). The flag leaf area showed significant associations with the grain yield in wheat (Monyo and Whittington, 1973). Chlorophyll is an important photosynthetic pigment to and largely determine photosynthetic capacity (Li et al., 2018). Chlorophyll fluorescence provides a signature of photosynthesis and can easily be measured with a portable chlorophyll fluorometers (Murchie and Lawson, 2013).

4)     Line 336-344: In a Rht-near-isogenic line study, the dwarf mutant Rht-B1c plants had thicker leaves, less leaf area and greater drought tolerance compared with wild allele Rht-B1a (Kocheva et al., 2014; Petrov et al., 2018). It has been suggested that a role for the Rht-B1c-encoded DELLA proteins provide some protective mechanisms in wheat (Dobrikova et al., 2017). We do not know if Rht-B1b also leads thicker leaves in Kauz (Zhang et al., 2013). Moreover, the 2nd leaf is important for carbon export, but we did not measure the area of these leaves. The current results showed that the higher flag leaf area in Westonia did not lead to higher biomass or grain weight. It has been augured that grain yield is poorly correlated with leaf photosynthetic rate when comparing across different genotypes (Long et al., 2006). However, the photosynthesis in ear parts showed significant contributions to wheat grain (Araus et al., 1993; Sanchez-Bragado et al., 2016).

5)     Line 347-349: Lines with low chlorophyll content were always associated with insufficient regulation of linear electron transport and a limited ability to prevent over-reduction of PSI acceptor side regardless of the genotype, environment, and growth stage (Zivcak et al., 2019).

6)     Additional 10 references were added, which include all these suggested by reviewer.

Araus JL, Brown HR, Febrero A, Bort J, Serret MD. 1993. Ear photosynthesis, carbon isotope discrimination and the contribution of respiratory CO2 to differences in grain mass in durum wheat. Plant Cell Environ. 16, 383-392.

Dobrikova AG, Yotsova EK, Börner A, Landjeva SP, Apostolova EL. 2017. The wheat mutant DELLA-encoding gene (Rht-B1c) affects plant photosynthetic responses to cadmium stress. Plant Physiol. Bioch. 114, 10-18.

Kocheva K, Nenova V, Karceva T, Petrov P, Georgiev GI, Börner A, Landjeva S. 2014. Changes in water status, membrane stability and antioxidant capacity of wheat seedlings carrying different rht-b1 dwarfing alleles under drought stress. J. Agron. Crop Sci. 200, 83-91.

Li Y, He N, Hou J, Xu L, Liu C, Zhang J, Wang Q, Zhang X, Wu X. 2018. Factors influencing leaf chlorophyll content in natural forests at the biome scale. Front. Ecol. Evol. 6.

Long SP, Zhu X-G, Naidu SL, Ort DR. 2006. Can improvement in photosynthesis increase crop yields? Plant Cell Environ. 29, 315-330.

Monyo JH, Whittington WJ. 1973. Genotypic differences in flag leaf area and their contribution to grain yield in wheat. Euphytica 22, 600-606.

Murchie EH, Lawson T. 2013. Chlorophyll fluorescence analysis: a guide to good practice and understanding some new applications. J. Exp. Bot. 64, 3983-3998.

Petrov P, Petrova A, Dimitrov I, Tashev T, Olsovska K, Brestic M, Misheva S. 2018. Relationships between leaf morpho-anatomy, water status and cell membrane stability in leaves of wheat seedlings subjected to severe soil drought. J. Agron. Crop Sci. 204, 219-227.

Sanchez-Bragado R, Molero G, Reynolds MP, Araus JL. 2016. Photosynthetic contribution of the ear to grain filling in wheat: a comparison of different methodologies for evaluation. J. Exp. Bot. 67, 2787-2798.

Zivcak M, Brestic M, Botyanszka L, Chen Y-E, Allakhverdiev SI. 2019. Phenotyping of isogenic chlorophyll-less bread and durum wheat mutant lines in relation to photoprotection and photosynthetic capacity. Photosynth. Res. 139, 239-251.

Reviewer 3 Report

This manuscript was clearly revised by the authors according to my previous review. I agree with the author’s explanations. However, I think the explanations should be described in the text.

2), 4) The authors should describe the necessary of comprehensive transcriptome analysis such as RNA-seq to understand whole genetic mechanisms to transportation of sucrose in the further studies at the Discussion section.

7) The authors should describe the explanation about the primer design for TaSUT1 genes at the Materials and Methods section such as L. 469-471.

Author Response

Review 3

This manuscript was clearly revised by the authors according to my previous review. I agree with the author’s explanations. However, I think the explanations should be described in the text.

2), 4) The authors should describe the necessary of comprehensive transcriptome analysis such as RNA-seq to understand whole genetic mechanisms to transportation of sucrose in the further studies at the Discussion section.

Response: In the discussion section, the following paragraph was changed to (line 381-394):

Line 381-394: During grain filling, significant differences in the sucrose levels of the leaves and stems appeared in Westonia and Kauz at 21 DAA - a critical stage of grain filling. Because of the closest and direct relationship between sucrose level and sucrose transporters, sucrose transporters were focused in our studies. In wheat, so far, the five SUT gene families have been studied, namely TaSUT1, TaSUT2, TaSUT3, TaSUT4, and TaSUT5. Compared with other SUT gene groups, TaSUT1 is more higher expressed in all parts of the wheat plant (Al-Sheikh Ahmed et al., 2018). The high gene expression levels (1000 to 2000 units) of three TaSUT1 homeologous genes, namely, TaSUT1_4A, TaSUT1_4B and TaSUT1_4D also showed in a RNAseq data set of a drought experiment. Therefore, TaSUT1 expression was focused using the same tissue samples for the sugar diurnal analysis at 21 DAA.

7) The authors should describe the explanation about the primer design for TaSUT1 genes at the Materials and Methods section such as L. 469-471.

Response: In materials and methods section, the following text was added in line 493-496:

Line 493-496: TaSUT1 gene primer pair used were based on the conserved region of TaSUT1_4A, TaSUT1_4B and TaSUT1_4D but different from the gene sequences of other four TaSUT gene groups (forward sequence: TGGATTCTGGCTCCTTGAC and reverse sequence: GCCATCCAAGAACAGAAGATT) (Al-Sheikh Ahmed et al. 2018).

Round 3

Reviewer 2 Report

The authors revised the manuscript according to the comments of the reviewer thoroughly and response the comments point by point, at present, the manuscript could be accepted.

Author Response

Thanks,

The introduction has been rewritten.

The whole text has been reviewed by a sugar specialist - Prof. Wim Van den Ende.